# Comprehensive Assessment of Graphene Oxide Nanoparticles: Effects on Liver Enzymes and Cardiovascular System in Animal Models and Skeletal Muscle Cells

**DOI:** 10.3390/nano14020188

**Published:** 2024-01-13

**Authors:** Milena Keremidarska-Markova, Iliyana Sazdova, Bilyana Ilieva, Milena Mishonova, Milena Shkodrova, Kamelia Hristova-Panusheva, Natalia Krasteva, Mariela Chichova

**Affiliations:** 1Faculty of Biology, Sofia University St. Kliment Ohridski, 1164 Sofia, Bulgaria; m_keremidarska@uni-sofia.bg (M.K.-M.); i.sazdova@biofac.uni-sofia.bg (I.S.); b.ilieva@biofac.uni-sofia.bg (B.I.); mmishonova@biofac.uni-sofia.bg (M.M.); mshkodrova@biofac.uni-sofia.bg (M.S.); 2Institute of Biophysics and Biomedical Engineering, Bulgarian Academy of Sciences, 1113 Sofia, Bulgaria; KPanusheva@biomed.bas.bg

**Keywords:** graphene nanomaterials, photothermal treatment, mitochondrial ATPase, polyamines, cardiac activity

## Abstract

The growing interest in graphene oxide (GO) for different biomedical applications requires thoroughly examining its safety. Therefore, there is an urgent need for reliable data on how GO nanoparticles affect healthy cells and organs. In the current work, we adopted a comprehensive approach to assess the influence of GO and its polyethylene glycol-modified form (GO-PEG) under near-infrared (NIR) exposure on several biological aspects. We evaluated the contractility of isolated frog hearts, the activity of two rat liver enzymes–mitochondrial ATPase and diamine oxidase (DAO), and the production of reactive oxygen species (ROS) in C2C12 skeletal muscle cells following direct exposure to GO nanoparticles. The aim was to study the influence of GO nanoparticles at multiple levels—organ; cellular; and subcellular—to provide a broader understanding of their effects. Our data demonstrated that GO and GO-PEG negatively affect heart contractility in frogs, inducing stronger arrhythmic contractions. They increased ROS production in C2C12 myoblasts, whose effects diminished after NIR irradiation. Both nanoparticles in the rat liver significantly stimulated DAO activity, with amplification of this effect after NIR irradiation. GO did not uncouple intact rat liver mitochondria but caused a concentration-dependent decline in ATPase activity in freeze/thaw mitochondria. This multifaceted investigation provides crucial insights into GOs potential for diverse implications in biological systems.

## 1. Introduction

Nanoparticles have rapidly integrated into our daily lives, such as in cosmetics, food packaging, therapeutics, biosensors, etc. [1]. As their applications expand, more people will come into contact with these materials. Despite their numerous benefits, many questions must be addressed before widespread application [2,3]. Extensive research has been conducted on the evaluation of the toxicity of nanoparticles. The data, however, is still controversial and confusing [4]. A comprehensive understanding of nanoparticles’ toxicological impact on the organism is still necessary [5].

Carbon-based nanomaterials are increasingly used in many scientific and technological fields owing to their exceptional mechanical, optical, and electrical properties [6]. Recently, graphene and its derivatives, such as graphene oxide (GO), have been identified as promising nanomaterials with a wide range of biomedical applications, including biosensors, cell and tumor imaging, adsorption of enzymes, and innovative applications like heart valve prostheses [7,8,9,10,11,12,13,14,15,16]. Recent investigations have revealed the significant potential of GO nanoparticles as carriers for various molecules [17,18,19]. Consequently, extensive research has been conducted on GO nanomaterials as smart drug delivery systems, representing a novel approach to combating malignant diseases [20,21,22]. 

Despite these promising characteristics, in vitro and in vivo cytotoxicity, genotoxicity, and GO nanomaterials’ underlying mechanisms of action have not been fully explored yet. Their potential adverse effects on living cells and organs are a limiting factor for medical applications [23]. Animal model studies have demonstrated that intravenously or intraperitoneally administered GO nanoparticles primarily accumulate in the lungs, liver, and spleen [24,25,26,27]. The severity of these toxic effects depends on the dosage, administration route, synthesis method, and physicochemical properties of GO [23]. GO and its derivatives interact with the cell membrane and can be internalized [28,29], accumulating in the cytosol and within organelles like lysosomes, mitochondria, endoplasm, and cell nuclei in a time- and dose-dependent manner [24,30,31,32,33]. The induction of cytotoxicity by GO involves plasma membrane damage and oxidative stress [34,35,36]. Various surface modifications of GO nanoparticles are under examination to enhance biocompatibility and efficiency as a drug carrier [37,38]. Polyethylene glycol (PEG) functionalization has shown an efficient reduction in the aggregation of different nanomaterials, their phagocytosis and opsonization by immune cells, and clot formation in the case of intravenous administration [39,40,41].

Recently, scientific interest has focused on combining photothermal therapy (PTT) and chemotherapy as a more efficient anti-cancer therapeutic approach [42,43]. Photothermal therapy uses a photothermal agent (PA) to convert light energy into heat. This generates high local temperatures that induce thermal ablation of tumor cells [44,45,46]. GO has demonstrated effectiveness as a PA for cancer PTT [47,48,49]. Moreover, GO functionalized with PEG and combined with near-infrared (NIR) irradiation has potential as a biocompatible smart nanocarrier in colon cancer cells [50]. Nevertheless, further studies at the organ and organism levels still need to understand its effects comprehensively for future biomedical applications.

Mitochondria are key organelles participating in vital biological processes [51,52,53,54,55,56,57] and frequently target sites targeted by nanomaterials [58,59]. Therefore, approaches based on targeting mitochondria hold great promise for cancer therapy [60]. Numerous studies have shown that in cancer cells, graphene and GO can physically damage mitochondria, reduce the mitochondrial membrane potential, alter the normal permeability of the mitochondrial membrane, and thus impair ATP production and cause reactive oxygen species (ROS) generation [28,30,34,55,61,62,63]. While these studies have found decreased levels and activities of proteins related to the respiration chain and suppressed ATP production [32,36,62], there is currently no documented evidence regarding the direct or indirect effects of graphene and its derivatives on the activity of mitochondrial ATPase–an enzyme playing a crucial role in ATP synthesis and breakdown within mitochondria. [64]. Therefore, in this work, we aimed to investigate the direct effects of graphene oxide on mitochondrial ATPase activity by using intact (coupled) mitochondria, and disrupted by freeze-thawing mitochondria.

We have also evaluated, for the first time, the activity of another important liver enzyme, diamine oxidase (DAO). It belongs to the semicarbazide-sensitive amine oxidases family (SSAO, EC 1.4.3.6) and is responsible for the conversion of primary amines to aldehydes, the production of hydrogen peroxide, and ammonia [65]. SSAOs are found in both eukaryotes and prokaryotes, including microorganisms, plants, animals, and humans. Their oxidase activity has been associated with various biological processes such as cell interactions, adipocyte differentiation, glucose transport, and vascular smooth muscle structural organization. Specifically, intracellular DAO catalyzes the deamination of primary diamines such as putrescine and histamine [65], which play a key role in cell proliferation and differentiation in physiological and tumor growth and inflammation [66]. Maintaining optimal levels of biogenic polyamines through DAO activity is crucial for normal cellular function.

Furthermore, in the context of the potential intravenous application of GO-based smart nanocarriers, exploring their impact on cardiovascular system activity is important but still elusive. Given that this system regulates vital physiological processes in the body, any alteration caused by a nanomaterial could have widespread effects on the patient’s overall health [67]. Some previous studies have shown that GO nanomaterials induce cardiotoxicity in cardiac cell lines after in vivo application in mice [68]. Still, the effects of PEG-functionalization and NIR-irradiation on GO nanoparticles have not been investigated yet.

Therefore, in this study, we aimed to explore the impact of GO and its polyethylene glycol-modified form combined with near-infrared (NIR) irradiation on several biological aspects at organ, cellular, and subcellular levels. For this, we have adopted different in vitro experiments to evaluate the effects of varying concentrations of GO nanoparticles (4, 10, 25, 50, and 100 μg/mL) on the contractility of isolated hearts from the frog Pelophylax ridibundus, ROS production in C2C12 skeletal muscle cells, and the activity of two rat liver enzymes (mitochondrial ATPase and DAO). This approach will allow us to better understand the effect of GO nanoparticles in normal living cells and their potential for different therapies, including chemotherapy and phototherapy for cancer.

## 2. Materials and Methods

### 2.1. Chemicals

Sucrose, potassium chloride (KCl), magnesium chloride (MgCl_2_), potassium dihydrogen phosphate (KH_2_PO_4_), sulfuric acid (H_2_SO_4_), sodium deoxycholate (C_24_H_39_NaO_4_), iron(II) sulfate heptahydrate (FeSO_4_.7H_2_O), ammonium molibdate (H_2_SO_4_(NH_4_)6Mo_7_O_24_.4H_2_O), and Folin reagent were purchased from Merck (Germany); sodium chloride (Na Cl), calcium chloride (CaCl_2_), sodium hydrogen carbonate (NaHCO_3_), N-2-hydroxyethylpiperazine-N′-2-ethanesulfonic acid (HEPES), semicarbazide, 4-aminoantipirine, putrescine, horseradish peroxidase, phenol, adenosine-5′-triphosphate disodium salt (ATP) and Tris(hydroxymethyl)aminomethane (Tris) were obtained from Sigma Aldrich (USA); sodium phosphate monobasic (NaHPO_4_.2H_2_O), potassium hydroxide (KOH) and sodium hydroxide (NaOH)—from Fluka (Switzerland); ethylene diamine tetraacetate disodium salt (EDTA), bovine serum albumin (BSA)—from Serva (USA); perchloric acid (HClO_4_)—from Riedel de Haën (Germany); 2,4-Dinitrophenol (DNP)—from BDH (England); disodium hydrogen phosphate (Na_2_HPO_4_.12 H_2_O)—from Valerus, Bulgaria.

Fresh stock solution of ATP were prepared prior to each experiment and kept on ice to preserve it from hydrolysis. Double distilled water was used for preparation of all reagent solutions.

### 2.2. Preparation and Characterization of PEGylated Graphene Oxide (GO-PEG)

Commercially purchased GO particles (Graphenea, San Sebastian, Spain) were diluted to a concentration of 1 mg/mL and sonicated for 2 h at 500 W using an ultrasonic homogenizer (VCX 500, Sonics and Materials, Inc., Newtown, CT, USA). The nanosized graphene oxide (GO) was modified with PEG following a previously established method [69]. Briefly, GO nanoparticles were mixed with mPEG-NH_2_ (Abbexa Ltd., Cambridge, UK), sonicated for 5 min, left overnight at 70 °C in a water bath, and the resultant GO-PEG suspension was centrifuged at 13,000× *g* for 20 min to remove any unstable aggregates and stored at 4 °C. 

For experimental purposes, particle suspensions were prepared as 1 mg/mL stock solutions or 0.4 mg/mL in distilled water. Before addition to the assay media, these suspensions were sonicated for 1 h in an ultrasonic water bath (50 Hz, UST2.4-150, Siel, Bulgaria). In the experiments involving NIR-irradiated GO and GO-PEG in animal models (rats and frogs), just before the assays, the nanoparticles were irradiated for 5 min at room temperature (GO-NIR and GO-PEG-NIR nanoparticles) using NIR-based source (laser) with peak emission of 808 nm (NIR region) and irradiance of 1.5 W/cm^2^.

The physicochemical characterization of GO and GO-PEG is described thoroughly in our previous publications [50]. The analysis involving the determination of particle size distributions, average particle size, and zeta potential of GO and GO-PEG nanoparticles was carried out using Dynamic Light Scattering (DLS) through the Zetasizer from Malvern Instrument, Ltd., Worcestershire, UK. The nanoparticles’ morphology was analyzed using a transmission electron microscope (TEM, JEM-2100, Tokyo, Japan). Additionally, the adsorption spectra of both nanoparticles in the UV-Vis and NIR regions were measured using a UV-Vis spectrophotometer (Specord 210 Plus, Edition 2010, Analytik Jena AG, Jena, Germany).

### 2.3. Animals

The livers used for assessing DAO and mitochondrial ATPase activities were obtained from male Wistar rats aged between 50 and 60 days. These rats were supplied by Vivarium Physiological Laboratory as part of the Project BG05M2OP0011.002-0012 CREATION AND DEVELOPMENT OF CENTRES OF COMPETENCE “Sustainable utilization of bio-resources and waste of medicinal and aromatic plants for innovative bioactive products” To investigate GO nanoparticles’ effects on the cardiovascular system, frogs of the species *Pelophylax ridibundus*, weighting approximately 40–50 g, were used. The frogs were supplied by the Vivarium of the Faculty of Biology at Sofia University “St. Kliment Ohridski” All experimental procedures were strictly conducted in accordance with the regulations outlined in Directive 2010/63/EU of the European Parliament and of the Council of 22 September 2010 regarding the protection of animals used for scientific purposes. Furthermore, the research was conducted under permit No. 224 from 23 January 2019, issued by the Bulgarian Food Safety Agency under the Ministry of Agriculture, Food, and Forestry. 

### 2.4. Isolation of Frog Hearts and Measurement of Heart Contractions

The frogs were denervated, and their hearts were cannulated and isolated. These excised heart preparations preserved functionally active sympathetic nerve connections [70]. This model allows the study of the effects of GO nanoparticles on both excitable tissues simultaneously—the heart muscle and the adrenergic axons. All experiments were performed at room temperature (20–22 °C). A cannula was inserted via the left aortic branch and aortic trunk and positioned into the ventricle. All solutions studied were administered through this cannula. The excised hearts were connected to a highly sensitive force transducer, GRASS FT03 (Grass Instrument Co., Quincy, MA, USA). The data were recorded and analyzed by TENZOSU software, version 1.3 (Stocks, Sofia, Bulgaria) as previously detailed [71]. 

The cardiac activity of untreated preparations (referred to as time controls) was measured at 15 min intervals with 0.2 mL of modified Ringer solution in the cannula. The composition of Ringer solution was as follows: 100 mM NaCl, 1.3 mM KCl, 0.7 mM CaCl_2_, 5 mM HEPES, and 1.2 mM NaHCO_3_, pH 7.0. During the initial 10 min of the experiment, spontaneous heart contractions diminished, followed by the development of regular contractions with stable patterns and force. Consequently, the force of the heart contractions at the 10th minute in each experiment was considered to be 100%, and the rest of the values were expressed as a percentage relative to this baseline. Each application of a fresh solution in the cannula caused a short-term increase in heart contractions, irrespective of the solution composition. In the experimental groups, following a 15 min adaptation period, GO or GO-PEG nanoparticles were applied at final concentrations of 4, 10, 25, 50, and 100 µg/mL, either without or after NIR irradiation. 

### 2.5. Cells and Cell Culture Experiments

The murine skeletal muscle myoblast C2C12 cells (CRL-1772, ATCC, LGC standard, Lomianki, Poland) were cultured in Dulbecco’s Modified Eagle’s Medium (DMEM; Sigma-Aldrich Chemie, Steinheim, Germany) containing 4.5 g/L glucose and supplemented with 10% fetal bovine serum (FBS, Thermo Fisher Scientific, Waltham, MA, USA), 50 U/mL penicillin, and 50 µg/mL streptomycin (Thermo Fisher Scientific). For the cell experiments, cells were seeded onto 24-well plates at a concentration of 3 × 10^4^ cells per well and incubated for 24 h. The next day, cells were exposed to the nanoparticles at a 100 μg/mL concentration for 24 h. The next day, the cells were irradiated with an NIR laser (at 808 nm) for 5 min and subjected to a DCFA-DA assay. 

### 2.6. DCFA-DA Analysis

The generation of hydrogen peroxide (H_2_O_2_) in C2C12 cells was determined using 2′,7′-dichlorofluorescin diacetate (DCFH-DA, Sigma-Aldrich). DCFH is a non-fluorescent compound that can permeate cell membranes. In the presence of cellular esterase and intracellular H_2_O_2_, it undergoes oxidation to form the fluorescent compound 2′,7′-dichlorofluorescein (DCF). Following the incubation of cells with nanoparticles, 20 µM DCFH-DA was added, and the cells were incubated at 37 °C and 5% CO_2_ for 30 min. Subsequently, the DCFH-DA-containing medium was removed, and the cells were rinsed with PBS before being scraped. The fluorescence intensity of DCF was measured using a spectrofluorometer with excitation at 485 nm and emission at 530 nm. Untreated cells were used as a control in these experiments.

### 2.7. Mitochondrial ATPase Activity Assay

The effect of GO nanoparticles on ATPase activity was studied using two distinct mitochondrial preparations: (i) intact mitochondria, also referred to as coupled mitochondria, and (ii) freeze-thawing disrupted mitochondria. The intact liver mitochondria were isolated following the procedure described by Chichova et al. [72]. These mitochondria were either used within 4 h of isolation or stored at a temperature of −15 °C to −20 °C. Following freezing/thawing, the mitochondria become uncoupled and can be used for an ATPase activity assay. The biuret reaction with bovine serum albumin (BSA) as a reference standard determined the mitochondrial protein content. 

ATPase activity was determined by measuring the Pi released from ATP. The reaction with freeze-thawed mitochondria was carried out in a 1 mL assay medium containing 200 mM sucrose, 10 mM KCl, 50 mM Tris-HCl, and 0.1 mM EDTA-KOH (pH 7.4). Different volumes of the GO nanoparticles’ stock solutions, without and after NIR-irradiation, were added to reach final concentrations of 4, 10, 25, 50, or 100 µg/mL. Following a 10 min pre-incubation of the mitochondria in the assay medium at 37 °C, the reaction was initiated by the addition of ATP at a final concentration of 1 mM, continued for 5 min at 37 °C, and terminated by adding 0.4 mL of 3 M perchloric acid. 

The ATPase reaction with intact mitochondria was carried out at room temperature with continuous stirring in a 4 mL assay solution. This solution consisted of 200 mM sucrose, 10 mM KCl, 50 mM Tris-HCl, 0.1 mM EDTA-KOH, 1 mM ATP (pH 7.4), and 50 µM 2.4-dinitrophenol (DNP) wherever indicated. GO nanoparticles from stock solutions were added both without and after NIR-irradiation to reach final concentrations of 100 µg/mL. The reaction was started by adding the mitochondrial suspension. Samples of 0.5 mL were extracted at intervals of 30, 60, 120, 180, 300, and 600 s of incubation and added to 0.2 mL of 3 M perchloric acid for termination of the reaction.

In all cases, the protein precipitates and nanoparticles were removed by centrifugation at 8800× *g* for 30 min. The concentration of Pi in the resulting supernatant was measured on a spectrophotometer S-22 UV/Vis, Boeco, Germany, at *λ* = 750 nm using the method of Fiscke and Subbarow [73] with some modifications. Parallel blanks in which the reaction was blocked by adding perchloric acid before ATP addition were prepared to determine the background Pi amount due to non-enzymatic hydrolysis. The mitochondrial ATPase activity was expressed as μmol Pi/mg protein/min for freeze-thawed mitochondria or as μmol Pi/mg protein for intact mitochondria.

### 2.8. Diamine Oxidase Activity Assay

The liver DAO was isolated, and its activity was measured using putrescine as a substrate according to the method of Dimitrov [74]. Following decapitation, the rat liver was quickly isolated, rinsed with cold sodium phosphate buffer (10 mM, pH 7.0), weighed, and then homogenized (1:4, *w*:*v*) with the same buffer. The resulting homogenate was heated at 60 °C for 10 min and centrifuged at 20,000× *g* for 20 min. The supernatant obtained after centrifugation was collected for further analysis. The protein content was determined using the method of Lowry et al. [75], with BSA as a standard.

The samples containing a reaction medium of 100 mM sodium phosphate buffer (pH 7.4), 4 IU/mL of horseradish peroxidase, and 0.3 mL of liver supernatant were pre-incubated at 37 °C for 20 min. In GO-treated samples, aliquots of the GO or GO-PEG nanoparticle stock solutions were added without or after NIR-irradiation to reach a final concentration of 4, 10, 25, 50, or 100 µg/mL. Blanks containing the same components with semicarbazide at a final concentration of 1 mM, included for DAO inactivation, were prepared to determine the pure DAO activity.

Following pre-incubation, the reaction was initiated by adding 2.5 mM putrescine, 0.82 mM 4-aminoantipirin, and 10.6 mM phenol (at final concentrations) and continued for 60 min at 37 °C. Termination of the reaction was carried out by cooling the tubes on ice and adding semicarbazide to the samples, reaching the final volume of 3 mL. Subsequently, nanoparticles were removed from the reaction mixture by centrifugation at 8800× *g* for 30 min. DAO activity was assessed spectrophotometrically at λ = 500 nm, measuring hydrogen peroxide produced in the amine oxidase reaction. The DAO activity was expressed as nmol H_2_O_2_/60 min/mg protein. 

### 2.9. Statistical Analysis

The data obtained are presented as mean values ± standard error of the mean (SEM). To assess the difference between the samples treated with nanoparticles and the untreated control of the independent samples, a Student’s *t*-test was employed when applicable. In cases where the Shapiro–Wilk test showed that the data were not normally distributed, the non-parametric Mann–Whitney Rank Sum test was performed. A value of *p* < 0.05 was considered significant. All statistical analyses were computed using SigmaPlot version 11.0.

## 3. Results

### 3.1. Physicochemical Characteristics of GO and GO-PEG Nanoparticles

After the preparation of the nanoparticles, their physicochemical properties were measured using various techniques to confirm the successful PEGylation of GO. TEM micrographs and Zeta Sizer measurements of GO nanoparticles revealed that the pristine GOs represent thin, transparent sheets with relatively smooth surfaces, a negative charge of −23.45 mV, and an average particle size of 261.7 nm (Figure 1a and Table 1). PEGylated GO nanoparticles exhibited larger dimensions, with an average particle size of 482.7 nm, a reduced negative charge of −13.63 mV, and a more wrinkled surface than pristine GO (Figure 1a and Table 1). This characteristic is advantageous for their functionalization with drugs or other bioactive molecules. We attribute the observed differences in size between the two types of GO nanoparticles to the larger PEG moiety (0.35 kDa) and the substitution of the negatively charged -COOH group in GO molecules with neutral PEG molecules, resulting in a lower negative ζ-potential. Both nanoparticle types demonstrated favorable absorbance within the NIR spectrum, notably at 808 nm, with the GO-PEG exhibiting higher NIR absorbance (Figure 1b).

Together, these results point to a successful GO modification with PEG and improved physicochemical properties such as aggregation, dispersity, hemocompatibility, etc., as demonstrated in our previous studies [41]. Our results are also in accordance with the results obtained by other authors, who have also shown the positive effect of PEGylation on different types of nanoparticles, including GO [39,40].

### 3.2. Effects of GO and GO-PEG without and after NIR-Irradiation on Isolated Frog Hearts

Considering the potential intravenous and pulmonary administration of GO-based nanomaterials, exploring their impact on cardiovascular system activity is essential because this system regulates crucial processes in the body. Therefore, we have evaluated the effects of GO at the organ level by using isolated frog hearts and determining their contractility after direct exposure to the increasing concentrations of the nanoparticles (4, 10, 25, 50, and 100 μg/mL) with and without NIR irradiation. Notably, GO nanoparticles increased the force of the heart contractions (Figure 2a, represented by filled circles), demonstrating statistical significance at 10, 25, and 50 μg/mL concentrations. Similarly, applying PEG-modified GO (Figure 2a, represented by filled triangles) elicited a notably significant positive inotropic effect even at the lowest concentration of 4 μg/mL. Interestingly, NIR irradiation of both types of nanoparticles, GO and GO-PEG (Figure 2b,c, respectively), did not remarkably alter the observed strength of the heart contractions. 

In all experimental groups, several fluctuations in the excitability and functionality of the cardiac conduction system were observed following the application of nanoparticles. This included reversible cardiac arrest, ventricular extrasystoles, and atrioventricular block (Figure 3). Importantly, the severity, duration, and frequency of these impairments increased with the concentration of applied nanoparticles. Such rhythm disturbances and heart rate variability could lead to a compensatory increase in the force of the heart contractions. Thus, the observed positive effect on heart contractility after nanoparticle treatment appears to be an indirect consequence of these excitatory disturbances serving as a compensatory mechanism.

In the frog heart, the cardiac conduction system includes a sinoatrial node located within the sinus venosus and an atrioventricular canal located in the atrial septum, which is analogous to the atrioventricular node in mammals. Unlike in mammalian hearts, there is no identified His-Purkinje network in the amphibian heart. Instead, ventricular trabeculae carry out this function, which serves as a high-speed conduction system for transmitting sinoatrial impulses across the ventricular wall [76]. In our model, the solution applied perfused only the ventricle. Therefore, the observed effects of nanoparticles were specifically related to these structures but not to sinoatrial and atrioventricular nodes in the heart. 

As mentioned above, GO has various applications in different fields, such as a contrast agent in human heart conduction system imaging with micro-computed tomography [77] and in the modeling of prosthetic heart valves [16]. However, it is crucial to note that GO exhibits potential adverse effects on biological systems at different levels. Zhang et al. [68] reported that GO-induced cardiotoxicity occurs through cell apoptosis, lactate dehydrogenase release, reduced mitochondrial membrane potential, ROS generation, and lipid peroxidation. Furthermore, research on zebrafish embryos exposed to GO in a concentration range of 0.1–1 mg/mL showed a significant increase in heart rate [78]. Our results support the suggestion that GO may cause some specific types of cardiac arrhythmias, like bradycardia or tachycardia, as an adverse effect. In our experimental model, where the applied nanoparticles exclusively perfused only the ventricle, their effect was limited to ventricular conduction structures. This limitation potentially impedes the propagation of physiological impulses from the sinoatrial node, enabling the ventricular equivalent of the His-Purkinje system to exhibit its automaticity.

Consequently, the observed data demonstrate a typical ventricular contractility pattern in response to both sinoatrial and ventricular impulses. This irregular cardiac function overloads the ventricle, causing excessive wall stretching and heightened contraction force. Our observations underscore the need for further investigation into the mechanisms underlying GO action on the heart’s conduction system to ensure its safe application in biomedical and engineering contexts. It is noteworthy that the spontaneous depolarization of His-Purkinje fibers results from an ionic current through “funny” (If) channels that operate on hyperpolarization to the diastolic voltage range [79]. Moreover, reports indicate that GO, especially GO-PEG, was adsorbed onto and/or penetrated the cell membrane of peritoneal macrophages, stimulating surface receptors [80]. Given the negative zeta potentials of GO (−23.45 mV) and GO-PEG (−13.63 mV) used in our study, there is a possibility that they might influence the cell membrane potential of both ventricular pacemaker cells and working myocardium, which can lead to decreased excitability and impaired conduction of impulses from the sinoatrial node. 

### 3.3. Effects of GO and GO-PEG without and after NIR-Irradiation on In Vitro ROS Production in Skeletal Myoblast C2C12 Cell Line Model

To evaluate the effects of the nanoparticles at the cellular level, GO nanoparticles were tested for possible cytotoxicity toward a skeletal muscle cell line, C2C12, by measuring the production of ROS after 24 h exposure to nanoparticles, with and without 5 min NIR irradiation. C2C12 murine myoblasts are the most widely used cell model for evaluating the interaction of various types of nanoparticles with muscle cells. The obtained results are presented in Figure 4. Among the two types of nanoparticles, GO-PEG induced lower ROS production than GO-treated cells. Interestingly, this suppressing ROS production effect was further amplified after NIR exposure. However, at the highest concentration (100 µg/mL), GO-PEG induced a 1.5-fold increase in ROS production, contrary to the general trend observed. Importantly, when compared to untreated control cells, GO and GO-PEG-treated cells triggered increased ROS generation, suggesting the potential toxicity of GO nanoparticles on skeletal muscles. Notably, even the smallest tested concentration of graphene oxide here could adversely impact muscle cells.

These findings contrast with some studies suggesting a positive influence of graphene oxide. For example, previous research demonstrated the ability of GO to reduce pro-inflammatory protein production in skeletal muscle cells exposed to SARS-CoV-2 spike protein [81]. These studies proposed that GOs might aid anti-inflammatory muscle therapy by scavenging proteins that trigger cytokine storms. However, the current findings highlight potential adverse effects, emphasizing the importance of further investigation into the precise mechanisms underlying GO nanoparticle interactions with skeletal muscle cells and their impact on cellular functions.

### 3.4. Effects of GO Nanoparticles on Rat Liver Mitochondria

As we mentioned above, the liver, lungs, and spleen are the primary sites/organs of accumulation of GO after intravenous or intraperitoneal administration. On the other hand, many studies have highlighted mitochondria as a primary organelle targeted by nanomaterials and potentially the primary site for the distribution of graphene and its derivates within the cells. Therefore, we have studied the activity of the liver mitochondrial enzyme ATPase after direct exposure to GO without and after NIR irradiation.

In previous experiments, we observed that 84.94 ± 2.04% (*p* = 0.014, *n* = 3) of the total ATPase activity of freeze-thawed mitochondria was sensitive to oligomycin. Olygomycin is recognized as a specific inhibitor of mitochondrial F-ATPase. We calculated the oligomycin-sensitive ATPase activity by determining the difference in the activity in the absence (total ATPase activity) and presence of oligomycin in the concentration of 1 μg/mg protein (oligomycin-insensitive ATPase activity). These results suggest that the predominant part of the measured mitochondrial ATPase activity was oligomycin-sensitive, and the contribution of other ATPases was negligible.

#### 3.4.1. Effects of GO and GO-NIR Nanoparticles on ATPase Activity of Intact Mitochondria

GO nanoparticles were initially tested for their potential to induce uncoupling in intact liver mitochondria. Figure 5 shows the results from two representative experiments with GO (Figure 5a) and GO-NIR (Figure 5b) nanoparticles at 100 µg/mL concentrations. ATPase activities remained consistently low during the 600 s registration under control conditions. However, adding the uncoupler DNP at a final concentration of 50 µM significantly enhanced ATP hydrolysis, indicating the normal functional status of the mitochondria and the low permeability of their inner membrane. 

Both GO and GO-NIR nanoparticles demonstrated no impact on the ATPase activity of intact mitochondria. Similarly, GO or GO-NIR nanoparticles have no significant effect on the ATPase activity of DNP-uncoupled mitochondria. These findings indicate that these nanoparticles do not possess a quick uncoupling effect on intact liver mitochondria and cannot pass readily through the inner mitochondrial membrane to influence ATPase activity.

The existing data highlights that graphene and GO physically damage the mitochondrial membrane, reducing mitochondrial membrane potential and suppressing ATP production [30,32,34,36,62,82]. In particular, GO treatment disrupted the normal mitochondrial permeability by continuously opening the mitochondrial permeability transition pore, leading to the uncoupling of oxidative phosphorylation [63]. However, it is important to note that all these observations were obtained in experiments on different cell lines with long exposures to graphene or GO ranging from 4 h to 96 h. The absence of an uncoupling effect of GO and GO-NIR nanoparticles on intact mitochondria in our experiments might be partially attributed to the short exposure (limited to the registration period of the ATPase reaction for 10 min). 

#### 3.4.2. Effects of GO and GO-NIR Nanoparticles on ATPase Activity of Freeze-Thawed Mitochondria

The freezing and thawing process disrupts the integrity of the mitochondrial inner membrane, resulting in uncoupling and stimulation of initial ATPase activity. Simultaneously, this disruption leads to increased permeability, partially allowing compounds from the surrounding medium to access the ATPase complex.

The ATPase activity values were calculated as percentages relative to the activity (expressed as µmol Pi/mg protein/min) measured under control conditions (in an assay medium free of nanoparticles). Exposure of mitochondria to GO nanoparticles at concentrations of 4, 10, 25, 50, and 100 µg/mL caused a concentration-dependent reduction in ATPase activity in freeze-thawed mitochondria (see Figure 6).

Experiments conducted with cell lines have revealed significant alterations in the levels and activities of respiration chain-related proteins after exposure to GO nanoparticles. GO administration increased mitochondrial oxygen consumption due to an increase in the activity of electron transport complexes I/III and an excessive supply of electrons to sites I/II of the electron transport chain [61]. Notably, a considerable decrease in complexes I, III, and IV activity in the NADH respiratory chain and a significant blockage of ATP synthesis after GO treatment in a concentration-dependent manner were observed following GO treatment [63]. Zhou and coworkers [32] also demonstrated that the exposure of cells to graphene suppressed the ATP production and decreased the activity of complexes I, II, III, and IV, but not that of complex V (ATP synthase), potentially due to interference with electron transfer between iron-sulfur centers. According to the authors, the lack of effect on the activity of complex V can potentially be due to the absence of an iron-sulfur center in this complex. However, our results showed a direct inhibition of the ATPase activity of freeze-thawed mitochondria exposed to GO nanoparticles. This observation contrasts the expected outcomes based on previous studies regarding the activities of various mitochondrial complexes following nanoparticle exposure.

In the case of GO nanoparticles, a significant impact was evident only at the highest concentrations of 100 µg/mL (reducing to 46.97 ± 6.59% of the control, *p* = 0.002; *n* = 6). However, when the nanoparticles were irradiated (GO-NIR, Figure 6) for 5 min before addition to the reaction medium, statistically significant inhibition was observed even at the lower concentrations (reducing to 81.32 ± 4.15%, 71.08 ± 3.22%, 61.21 ± 5.10%, and 46.39 ± 8.26% of the control at concentrations of 10, 25, 50, and 100 µg/mL, respectively; *p* = 0.002; *n* = 6). These findings suggest that NIR irradiation might enhance the GO nanoparticle effects on the ATPase activity of rat liver mitochondria. 

NIR irradiation has shown the potential to amplify various aspects of nanodrug behavior like drug release, cellular uptake, cytotoxicity, and movement of nanodrugs into the cell nucleus [83]. In specific cancer cell studies, GO and NIR irradiation demonstrated a potent effect of 42% inhibition of mitochondrial activity after 24 h [69]. Zeng and coworkers [84] have shown a graphene-based approach that synergizes photodynamic and photothermal therapy. Their single-dose NIR irradiation (for 5 min at 808 nm) inhibited ATP synthesis and disturbed mitochondrial function and the energy supply in cancer cells. Moreover, Hu et al. [85] have observed that cancer cells treated with GO-based nanocomposites in combination with a low-temperature PTT exhibited mitochondrial depolarization, depletion of ATP, and substantial ROS generation, leading to mitochondrial-associated apoptosis. The formation of intracellular ROS triggered by GO after NIR irradiation is also demonstrated in mouse melanoma tumors by Kalluru et al. [86]. These studies demonstrate NIR irradiation’s ability to enhance drug effects, disrupt mitochondrial function, induce apoptosis, and generate ROS, particularly in cancer cells treated with graphene-based materials.

### 3.5. Effects of GO Nanoparticles on Rat Liver DAO Activity

#### 3.5.1. Effect of PEG-Modification on Rat Liver DAO Activity

The effects of various concentrations of GO and GO-PEG (4, 10, 25, 50, and 100 µg/mL) on DAO activity were studied in vitro using the enzyme extracted from rat liver homogenate supernatant. DAO activity was calculated as percentages relative to the activity (expressed as nmol/H_2_O_2_/60 min/mg protein) measured under control conditions (in a GO-free assay medium).

Interestingly, GO nanoparticles in concentrations of 10, 25, 50, and 100 µg/mL significantly stimulated DAO activity (reaching 155.05 ± 11.84% of the control, *p* = 0.013; *n* = 6) (see Figure 7). The functionalization with PEG enhanced the effects of GO nanoparticles on DAO activity, leading to a significant concentration-dependent increase, rising to 179.24 ± 13.27 at 100 µg/mL (*p* = 0.003, *n* = 6).

Higher DAO activity was registered in the presence of GO-PEG nanoparticles compared to GO in all studied concentrations. However, these differences between the groups did not reach statistical significance.

#### 3.5.2. Effects of NIR-Irradiation of GO and GO-PEG Nanoparticles on Rat Liver DAO Activity

The NIR-irradiated GO nanoparticles strongly stimulated DAO activity (Figure 8a). The most notable effects were at concentrations of 25, 50, and 100 µg/mL (270.69 ± 25.26, 300.93 ± 30.50, and 288.81 ± 24.72 compared to the control; *p* < 0.001). Similarly, a concentration-dependent stimulation of DAO activity was evident in samples with NIR-irradiated GO-PEG nanoparticles (Figure 8b), showing a significant increase in the activity compared to the control, reaching 273.62 ± 24.93 at 100 µg/mL (*p* < 0.001, *n* = 6). In most studied concentrations, NIR irradiation significantly enhanced the effects of both unmodified and PEG-modified GO nanoparticles on DAO activity (Figure 8).

Polyamines are essential for the survival of both normal and tumor cells, and their levels are dramatically elevated in tumor cells compared with normal cells. Dysregulated polyamine biosynthesis correlates with several pathological states [87,88]. In these cells, abnormally high levels of polyamines play a pivotal role in cellular proliferation, gene expression, and autophagic processes contributing to tumor progression. One plausible hypothesis is that activating DAO–an enzyme of the catabolic pathway- GO and GO-PEG nanoparticles could reduce the cell polyamine pool and thus suppress tumor growth. This potential antitumor effect might be further enhanced if the nanoparticles are exposed to NIR light before application.

## 4. Conclusions

The present comprehensive research on the effects of GO nanoparticles on various biological systems provides an in-depth insight into their impact. The GO nanoparticles studied in this work displayed various effects on different biological systems. They influenced the activity of the rat liver enzymes ATPase and DAO and induced the production of ROS in the skeletal muscle C2C12 cell line. They caused stronger contractions in isolated frog hearts, but they were arrhythmic with exciting disturbances.

The effects of GO nanoparticles on mitochondrial ATPase activity differed between intact and freeze-thawed mitochondria. Intact mitochondria showed no noticeable alteration in ATPase activity, even at a 100 μg/mL concentration. This suggests that these nanoparticles do not have a quick uncoupling effect on intact liver mitochondria. However, in freeze-thawed mitochondria, all tested concentrations of GO nanoparticles significantly inhibited ATPase activity in a concentration-dependent manner. NIR irradiation intensified this inhibitory effect, affecting ATPase activity even at lower concentrations. 

Moreover, GO and GO-PEG nanoparticles showed intriguing stimulation of DAO activity, indicating a potential to modulate polyamine levels, suggesting potential for tumor growth suppression. All concentrations of the nanoparticles studied induced a concentration-dependent increase in DAO activity compared to the control, with greater activation registered for the GO-PEG particles. NIR irradiation notably strengthened this effect, elevating DAO activity to 300% of the control levels. 

However, these nanoparticles also demonstrated notable effects on excised frog hearts, causing excitatory disturbances and rhythm irregularities. While the nanoparticles induced positive inotropic effects, they also triggered cardiac rhythm disturbances, indicating a complex impact on the cardiac conduction system. Surprisingly, irradiation with an NIR laser did not significantly influence the nanoparticles’ effects on the contractility of the frog hearts.

In skeletal muscle cell models, GO and GO-PEG nanoparticles increased ROS production, potentially indicating cytotoxicity towards muscle cells. This contrasted with some previous studies suggesting potential anti-inflammatory effects of GO in muscle cells, highlighting the need for a deeper understanding of these nanoparticles’ interactions with different cell types.

The outcomes emphasize the dual nature of GO nanoparticles–displaying both promising stimulatory effects on certain enzymatic activities like DAO and potential adverse impacts on mitochondrial function, cardiac rhythm, and ROS generation. These conflicting outcomes underscore the necessity for further exploration into the precise mechanisms underlying GO nanoparticle interactions with various biological systems and their implications for biomedical applications.

## Figures and Tables

**Figure 1 nanomaterials-14-00188-f001:**
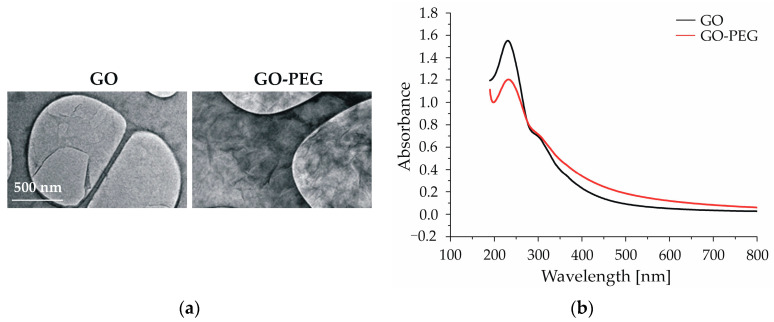
Physiochemical properties of GO and GO-PEG NPs: (**a**) TEM analysis of GO and GO-PEG after sonication; (**b**) UV-Vis spectra of GO and GO-PEG.

**Figure 2 nanomaterials-14-00188-f002:**
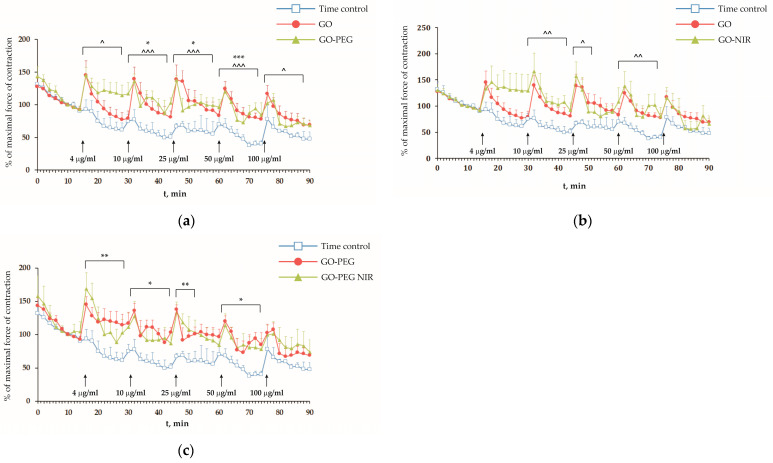
Effects of GO and GO-PEG without and after NIR-irradiation on excised frog hearts. The maximal force of the heart contractions in control conditions (nanoparticle-free medium) is shown for comparison (time control, (**a**–**c**), open squares). Arrows point the solution changes (fresh Ringer solution for time control and the corresponding nanoparticle concentrations for experimental groups). Data are plotted as mean ± SEM (*n* = 6). Asterisks indicate significant difference between GO (**a**) and GO-PEG (**c**) versus the corresponding time control at each minute of the experiment. Carets indicate significant differences between GO-PEG (panel a) and GO-NIR (**b**) versus the corresponding time control: * *p* < 0.05, ** *p* < 0.01, *** *p* < 0.001; ^ < 0.05, ^^ < 0.01, ^^^ < 0.001. The square brackets represent the time interval to which the observed significant deference refers.

**Figure 3 nanomaterials-14-00188-f003:**
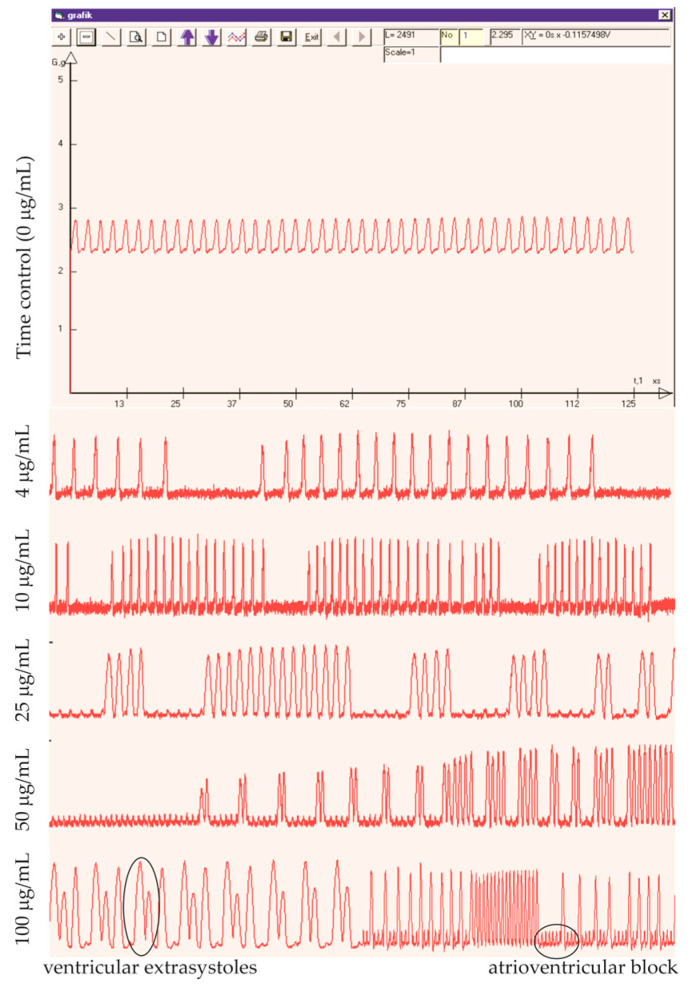
Representative original record of excised frog heart contractions at control conditions and after application of increasing concentrations of nanoparticles. Circles point observed deviations in excitability of the cardiac muscle.

**Figure 4 nanomaterials-14-00188-f004:**
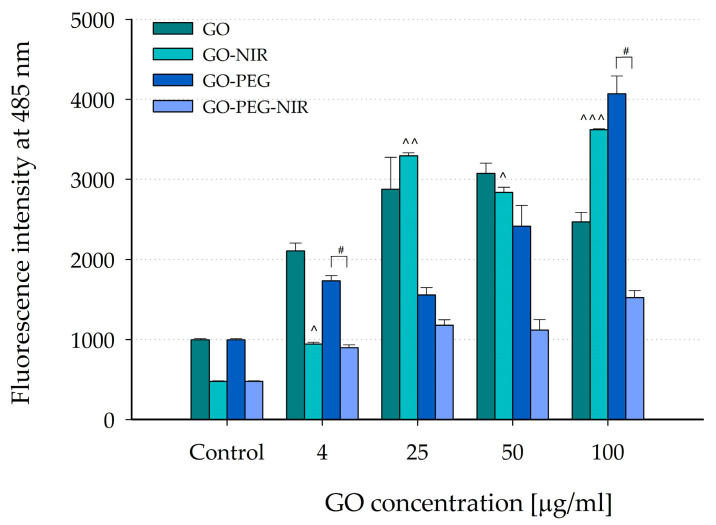
Effects of NIR irradiation of GO and GO-PEG nanoparticles on ROS levels. Data are presented as mean ± SEM of two independent experiments. Carets indicate significant differences for GO-NIR from the control: ^ *p* < 0.05, ^^ *p* < 0.01, ^^^ *p* < 0.001. Sharps indicate significant differences for GO-PEG-NIR from GO-PEG: ^#^ *p* < 0.05.

**Figure 5 nanomaterials-14-00188-f005:**
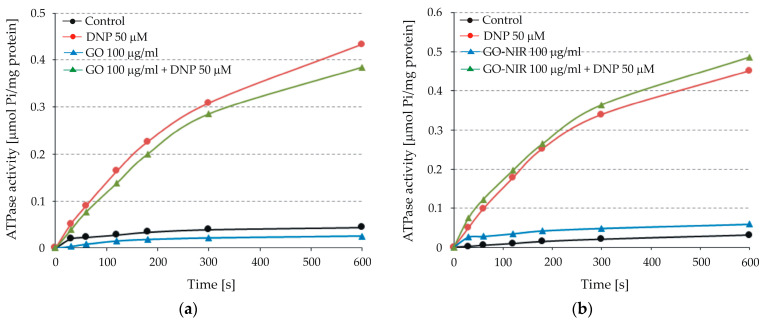
Effects of graphene oxide (GO) (**a**) nanoparticles and irradiated GO nanoparticles (GO-NIR) (**b**) on ATPase activity in both intact and 2.4-dinitrophenol (DNP)-uncoupled mitochondria The reactions were started by adding 80 µL of mitochondrial suspension (protein 7.22 mg/sample and 6.64 mg/sample for GO and GO-NIR nanoparticles, respectively) and conducted for 600 s as described in the Materials and Methods section. The data presents curves registered during a single experiment with GO and GO-NIR nanoparticles, respectively.

**Figure 6 nanomaterials-14-00188-f006:**
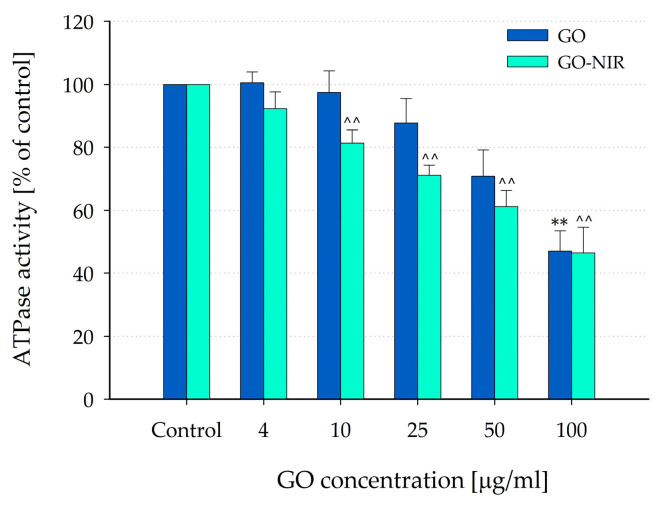
Effects of GO and GO-NIR nanoparticles on ATPase activity of freeze-thawed mitochondria. ATPase activity values were calculated as percentages of the activity measured under control conditions (nanoparticle-free assay media). Data are plotted as mean ± SEM of six independent experiments (three parallel samples per group per experiment). Asterisks and carets indicate significant differences from the control: ** *p* < 0.01 (for GO), ^^ *p* < 0.01 (for GO-NIR).

**Figure 7 nanomaterials-14-00188-f007:**
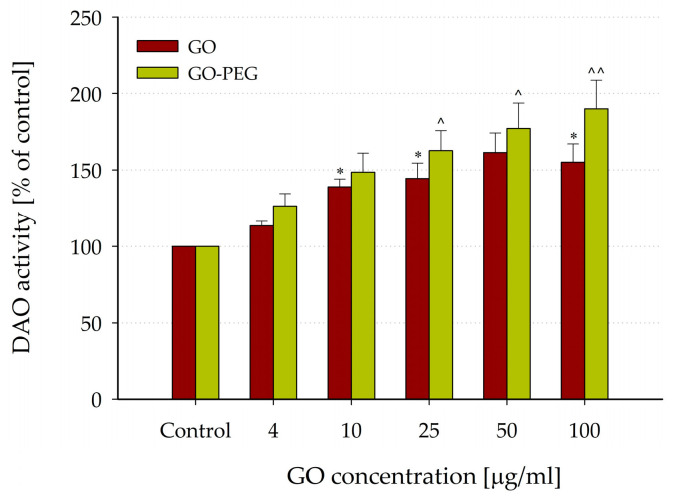
Effect of GO and PEG-modified (GO-PEG) nanoparticles on DAO activity. DAO activity values were calculated as percentages of the enzyme activity measured under control conditions (nanoparticles-free media). Data are presented as mean ± SEM of six independent experiments. Asterisks and carets indicate significant differences from the control: * *p* < 0.05 (for GO); ^ *p* < 0.05, ^^ *p* < 0.01 (for GO-PEG).

**Figure 8 nanomaterials-14-00188-f008:**
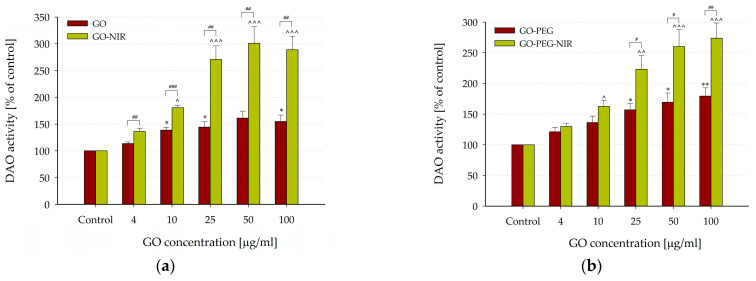
Effects of NIR irradiation of GO (**a**) and GO-PEG (**b**) nanoparticles on DAO activity. DAO activity values were calculated as a percentage of the enzyme activity measured under control conditions. Data are presented as mean ± SEM of six (seven for GO-NIR) independent experiments. Asterisks and carets indicate significant differences from the respective controls: * *p* < 0.05, ** *p* < 0.01 (for GO and GO-PEG); ^ *p* < 0.05, ^^ *p* < 0.01, ^^^ *p* < 0.001 (for GO-NIR and GO-PEG-NIR). Sharps indicate significant differences for GO-NIR and GO-PEG-NIR from GO and GO-PEG, respectively: ^#^ *p* < 0.05, ^##^ *p* < 0.01, ^###^ *p* < 0.001.

**Table 1 nanomaterials-14-00188-t001:** The average hydrodynamic size and zeta potential of GO and GO-PEG.

Parameters/Samples	GO	GO-PEG
Average particle size [nm]	261.7	482.7
Surface charge [mV]	−23.45	−13.63

## Data Availability

The data supporting this study’s findings are available from the corresponding authors, N.K. and M.C., upon reasonable request.

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
