# Peer review of "Comprehensive Assessment of Graphene Oxide Nanoparticles: Effects on Liver Enzymes and Cardiovascular System in Animal Models and Skeletal Muscle Cells"

_nanomaterials, 2024, doi:10.3390/nano14020188_

Round 1

Reviewer 1 Report

Comments and Suggestions for Authors

In this article, the authors employed a comprehensive approach to evaluate the effects of GO and its polyethylene glycol modified form (GO-PEG) on several biological aspects under near-infrared (NIR) exposure. The authors evaluated the contractility of isolated frog hearts, the activity of two rat liver enzymes - mitochondrial ATPase and diamine oxidase (DAO), and the production of reactive oxygen species (ROS) in C2C12 skeletal muscle cells after direct exposure to GO nanoparticles. This multifaceted investigation provides crucial insights into GO's potential for diverse implications in biological systems. I believe that publication of the manuscript may be considered only after the following issues have been resolved.

1.    In order to highlight the advantages of this work, it is recommended that the author supplement a comparative table of related work.

2.    To prove the existence of graphene, the authors need to make necessary judgments, such as Raman testing.

3.    Figure 3 suggests that the author use professional drawing software to process the relevant data, rather than simply taking screenshots.

4.    What is the physical mechanism by which graphene enhances related properties? Suggest authors to supplement relevant band analysis.

5.    Regarding the Graphene, some of the latest relaevant literature authors need to mention, such as Materials Research Bulletin 171 (2024) 112635; Opto-Electron Adv 5, 200098 (2022); Diamond & Related Materials 140 (2023) 110481; Opto-Electron Sci 2, 230012 (2023).

6.    The English expression of the whole article needs to be further improved.

Comments on the Quality of English Language

Minor editing of English language required

Author Response

Dear Reviewer,

Thank you very much for providing valuable feedback on our review paper.

We have addressed all your comments adequately and have provided detailed responses to all of them. Please, see below:

Comments and Suggestions for Authors

  1. In order to highlight the advantages of this work, it is recommended that the author supplement a comparative table of related work.

Response: We appreciate this suggestion. It is true that our experiments with respect to the evaluation of DAO activity and the direct effect of GO on ATPase have been conducted for the first time. Specifically, the innovative inclusion of NIR irradiation together with GO and GO-PEG nanoparticles is a notable highlight of our research. This aspect deserves emphasis within our paper as it underscores the pioneering nature of our investigations. Concerning the preparation of a comparative table of related work, there is not much comparable data regarding GO nanoparticle experiments in the literature, and such a table might be relatively short. Therefore, we have focused on discussing the literature findings in our Introduction section. This approach allows us to provide a comprehensive contextual background, establishing a solid foundation for understanding our contributions despite the limited tabulated data resulting from the scarcity of analogous experiments in the field.

  1. To prove the existence of graphene, the authors need to make necessary judgments, such as Raman testing.

Response: Thank you for bringing this to our attention. In our previous publications by Kamenska, T. et al.. (doi.org/10.3390/ma14174853), and Georgieva М. et al.,2021(doi: 10.3390/pharmaceutics13030424) a comprehensive assessment of the physicochemical properties of GO and GO-PEG was conducted and presented. We have used a diverse array of techniques, including Raman spectroscopy, FTIR (Fourier-transform infrared spectroscopy), Zeta sizer analysis, AFM (Atomic Force Microscopy), UV-Vis (Ultraviolet-visible spectroscopy), etc. Notably, the results derived from Raman measurements showed that the spectra of both nGO (graphene oxide) and nGO-PEG exhibited an intense G peak, pointing the preservation of the hexagonal aromatic rings typical for the graphene nanosheets. An intriguing observation was made after comparing the ID/IG ratio between nGO-PEG (0.927) and nGO (0.947) spectra. It suggested that by synthesizing nGO-PEG, there was an apparent restoration of the aromatic structures by rectifying defects that might have been present initially. This observation suggests a potential recovery and enhancement of the aromatic configuration during the nGO-PEG synthesis process, underscoring a meticulous repair mechanism employed in the synthesis protocol.

  1. Figure 3 suggests that the author use professional drawing software to process the relevant data, rather than simply taking screenshots.

Response: Thank you for your observation regarding Figure 3. We understand the need to implement professional drawing software for data processing rather than solely depending on screenshots. Our current tool, the TENZOSU software, is specialized in registering and analyzing cardiac contractions in a 90-minute experiment and produces comprehensive records. However, due to its limitations in exporting suitable graphics, we have resorted to screenshots, which can be further refined using image editing software to meet our requirements. In the revised version of our manuscript, following your and reviewer 3 recommendations we have substituted Fig. 3 with another one, demonstrating the original record of excised frog heart contractions under control conditions and subsequent alterations following the application of increasing nanoparticle concentrations. The image was further processed using an image editing program.

  1. What is the physical mechanism by which graphene enhances related properties? Suggest authors to supplement relevant band analysis.

Response: Thank you for your insightful query regarding the physical mechanisms through which graphene enhances associated properties. We acknowledge the importance of band analysis in supplementing and elucidating these mechanisms. Unfortunately, we haven't conducted such an analysis in our current study and cannot fulfill this request for now. However, we greatly appreciate your suggestion and understand the value it could add to comprehensively understanding the enhancements linked to graphene. In future research, we will incorporate band analysis to explore and clarify the physical mechanisms underlying the augmentation of properties facilitated by graphene. Your suggestion will undoubtedly guide our next investigations, and we aim to address this aspect in our future work. Thank you once again for your valuable input.

  1. Regarding the Graphene, some of the latest relaevant literature authors need to mention, such as Materials Research Bulletin 171 (2024) 112635; Opto-Electron Adv 5, 200098 (2022); Diamond & Related Materials 140 (2023) 110481; Opto-Electron Sci 2, 230012 (2023).

Response: in the revised version under reference number 12, 13, 14 and 15.

  1. The English expression of the whole article needs to be further improved.

Response: Thank you for highlighting this aspect. As we acknowledge the importance of ensuring clear and concise language in our work we have put additional efforts into improving overall English expression throughout the manuscript.

Thank you again for your feedback, and we hope these changes improve the quality of the manuscript.

Sincerely,

Prof. Natalia Krasteva

Assoc. Prof. Mariela Chichova

Reviewer 2 Report

Comments and Suggestions for Authors

nanomaterials-2809327

Recommendation: Publish in Nanomaterials after minor revision

In the current work, the authors used an integrated approach to evaluate the effects of GO and its polyethylene glycol modified form (GO-PEG) under near infrared (NIR) irradiation on several biological aspects: effects on liver enzymes and cardiovascular system in animal models and on skeletal muscle cells.

         Recent studies show that GO nanoparticles can be used in medicinal chemistry.   Extensive studies on GO nanomaterials can be considered as potential active agents in smart drug delivery systems, which represents a new approach to combat malignant diseases. Despite these promising properties, the cytotoxicity and underlying mechanisms of action of GO nanomaterials in vitro and in vivo are not fully understood. Their potential adverse effects on living cells and organs are a limiting factor for medical applications. Therefore, research in this area is extremely relevant today.

Minor points:

1.         The authors should include more recent updates on the topic and compare how this study advances current knowledge in the Introduction section. I found the introduction very long. I recommend shortening it a little.

2.         The "Conclusion" section should be more detailed.

Author Response

Dear Reviewer,

Thank you very much for providing valuable feedback on our review paper.

We have addressed all your comments adequately and have provided detailed responses to all of them. Please see below:

 Minor points:

 The authors should include more recent updates on the topic and compare how this study advances current knowledge in the Introduction section. I found the introduction very long. I recommend shortening it a little.

Response: Thank you for your comment on the Introduction section. We acknowledge that the introduction might be lengthy, and have tried to shorten it without compromising the logic of our study, as you can see in the revised version of the manuscript. 

We completely understand the importance of including more recent updates in the Introduction section to emphasize how our study advances current knowledge in graphene oxide experiments. However, despite our thorough search and the literature, we have encountered a challenge in sourcing up-to-date data, analogous to our experimental models. Indeed, our experiments concerning the evaluation of DAO activity and the direct effect of GO on ATPase have been conducted for the first time. The scarcity of recent publications about this specific area of research has limited our ability to include a broader spectrum of recent findings for comparative analysis in the Introduction section.  Therefore, in the revised version of our manuscript, in the Introduction Section, we have included additionally only several publications about the newest applications of graphene following the first reviewer's suggestions.

  1. The "Conclusion" section should be more detailed.

Response: Thank you for recommending enhancing the detail level within the "Conclusion" section. To enrich the clarity and depth of the "Conclusion," we have expanded and revised it in the updated version of the manuscript. You will find an extended "Conclusions" section in the revised manuscript that addresses your feedback.

Thank you again for your feedback, and we hope these changes improve the quality of the manuscript.

Sincerely,

Prof. Natalia Krasteva

Assoc. Prof. Mariela Chichova

Reviewer 3 Report

Comments and Suggestions for Authors

The manuscript entitled "Comprehensive Assessment of Graphene Oxide Nanoparticles: Effects on Liver Enzymes and Cardiovascular System in Animal Models and Skeletal Muscle Cells" evaluated cardiac contractility, the activity of two rat liver enzymes, mitochondrial ATPase and DAO, as well as activity and ROS production in C2C12 skeletal muscle cells after direct exposure to GO nanoparticles. We see that there's been a lot of work done on this by the researchers. However, many concerns should be addressed before it can be considered for publication.

1.We can see that the author detects physiochemical properties of GO and GO-PEG NPs. But in subsequent experiments, the authors also used the NIR-irradiated GO and GO-PEG. So do the physicochemical properties of NIR-irradiated particles of GO and GO-PEG change? We suggest the authors to complement the examination of the physicochemical properties of GO and GO-PEG after NIR irradiation. If it is not possible to supplement the experiments, we hope that the authors will give the basis that the physicochemical properties of GO and GO-PEG will not be changed after NIR irradiation.

2. How the authors determined the concentration of GO nanoparticles?

3. In Line 409, we can see that the authors elaborate that the liver, lungs and spleen are the main organs where GO particles accumulate after intravenous or intraperitoneal injection. But we don't see in the article that the authors elaborate on how GO particles are exposed to rats, could the authors please add the methodology.

4.In Figure 2, what does time control mean? Is it not treated with GO particles?

5. In the three graphs of Figure 2, we only see these concentrations of 4, 10, 25, 50, and 100 μg/mL, and whether or not the authors compared these five concentrations to 0 μg/mL when comparing their results. In the graph, the author's arrow is pointing to the peak of the line graph, so are the authors comparing just the size of that peak? Or is it a segment of the folded graph. We don't think the folded graph presents the authors' results well.

6.In Line 331, The authors indicated that the severity, duration, and frequency of these injuries increased with increasing concentrations of applied nanoparticles. But in Figure 3, we do not see the above expression. We hope that the authors can complete the figure by adding exactly which particles and how many concentrations were exposed, so that the reading public can understand the authors' results more clearly.

7.In Line 383, what is the basis for the authors' choice of 5 minutes of NIR irradiation?

8.In Figure 4, Line 385, we see in the description of the results that the authors compare the amount of ROS after GO and GO-PEG treatment. We think that in order to better present the authors' results, these four sets of data should be aggregated into a bar graph, if the authors want to compare them with each other.

9. In Line 380, the authors confirmed whether GO nanoparticles cause cytotoxicity by testing ROS levels, which we believe is incomplete. A complementary cell viability assay would have better interpreted the cytotoxicity caused by GO nanoparticles.

10. In the authors' figures, we can see that sometimes Control (Figure 5) is used to express the untreated group and sometimes 0 μg/mL (Figure 4) is used. If it is possible to standardize one form of presentation, it may make it easier for the reader to understand.

11.In Figure 5, authors didn’t detect ATPase activity of GO-PEG treated. If this part is added to make the author's results more complete.

12. The authors used frogs and rats as animal models respectively, why did the authors use two different animal models and on what basis?

Comments on the Quality of English Language

Minor editing of English language required.

Author Response

Dear Reviewer,

Thank you very much for providing valuable feedback on our review paper.

We have addressed all your comments adequately and have provided detailed responses to all of them.

Please, see below:

1.We can see that the author detects physiochemical properties of GO and GO-PEG NPs. But in subsequent experiments, the authors also used the NIR-irradiated GO and GO-PEG. So do the physicochemical properties of NIR-irradiated particles of GO and GO-PEG change? We suggest the authors to complement the examination of the physicochemical properties of GO and GO-PEG after NIR irradiation. If it is not possible to supplement the experiments, we hope that the authors will give the basis that the physicochemical properties of GO and GO-PEG will not be changed after NIR irradiation.

Response: Thank you for your insightful comment. While our experiments focused on the initial characterization of these nanoparticles, your suggestion to investigate potential changes in their properties following NIR exposure holds significant value. To meet your suggestion, we conducted further experiments and found that NIR irradiation did not cause any significant alterations in the assessed physicochemical characteristics of GO and GO-PEG. Parameters like average nanoparticle size, zeta potential, and UV-Vis adsorption change slightly. There was a minimal shift of only 1-3 mV in the zeta potential and 1-17 nm in average size after NIR irradiation of GO and GO-PEG nanoparticles, respectively,  and slightly decrease in the adsorption properties after NIR irradiation. You can see the detailed results of our experiments below.

Parameters/Samples

nGO before NIR

nGO after NIR

nGO-PEG before NIR

nGO-PEG  after NIR

Average particle size [nm]

245.114 ± 0.45

262.166 ± 6.7

296 ± 12.92

297.033 ± 6.87

Surface charge [mV]

-38.03 ± 0.56

-35.516 ±0.45

-22.68 ± 0.37

-21.86 ± 0.52

  1. How the authors determined the concentration of GO nanoparticles?

Response: Concerning graphene oxide, it is a ready-for-used solution obtained from Graphenea, Spain, with a 4 mg/ml concentration. Before our experiments, we diluted GO to the concentration of 1 mg/ml and obtained a stock solution further diluted to the working concentrations. Concerning PEGylated GO (GO-PEG), its concentration was determined by combining several techniques after synthesis. We used UV-Vis spectroscopy and dynamic light scattering (DLS) to measure absorbance, particle size, and surface characteristics as surface charge, which allowed us to accurately assess and quantify the concentration of GO-PEG nanoparticles in our experimental solutions.

  1. In Line 409, we can see that the authors elaborate that the liver, lungs and spleen are the main organs where GO particles accumulate after intravenous or intraperitoneal injection. But we don't see in the article that the authors elaborate on how GO particles are exposed to rats, could the authors please add the methodology.

Response: The sources cited in the manuscript – "Introduction" Section (line 52-54, References [24-27]), include in vivo studies with intravenously or intraperitoneally administered GO nanoparticles in animal models, while our study primarily focused on in vitro experiments assessing the effects of GO nanoparticles on rat liver mitochondrial ATPase and DAO. As we understand the importance of providing clarity on the exposure methodology, we have described it in detail and cited it correctly in the manuscript. Specifically, in the "Materials and Methods" section, subsections 2.6. "Mitochondrial ATPase Activity Assay" and 2.7. "Diamine Oxidase Activity Assay," we have highlighted that the GO nanoparticles, both before and after NIR-irradiation, were directly introduced into the assay media to achieve the desired final concentrations for the enzyme reactions, as follows: “Different volumes of the GO nanoparticles’ stock solutions, without and after NIR-irradiation, were added to reach final concentrations of 4, 10, 25, 50, or 100 µg/ml. Following a 10-minute pre-incubation of the mitochondria in the assay medium at 37°C, the reaction was initiated by the addition of ATP ….” – lines 249-251; “GO nanoparticles from stock solutions were added both without and after NIR-irradiation to reach final concentrations of 100 µg/ml. The reaction was started by adding the mitochondrial suspension…” – lines 257-259; “Following decapitation, the rat liver was quickly isolated, and …“ – lines 272-273; “In GO-treated samples, aliquots of the GO or GO-PEG nanoparticles stock solutions were added without or after NIR-irradiation to reach a final concentration of 4, 10, 25, 50, or 100 µg/ml.” – lines 280-282.

  1. In Figure 2, what does time control mean? Is it not treated with GO particles?

Response: The time controls represent frog hearts in which the cardiac activity was registered for 90 minutes in standard conditions without treatment (nanoparticles-free Ringer solution in the cannula). The term "time" reflects changes in heart activity that occur during the 90-minute recording of the heart contractions and are not due to any additional treatment. The registration of time controls was necessary because, in general, a slight decrease in the contractions' force was observed during the experiment. In addition, every replacement of the solution in the cannula caused a short-term increase of the contraction force, which was in accordance with the Frank-Starling mechanism. In time controls, every 15 minutes the Ringer solution in the cannula was replaced with a fresh one in order to maintain the heart intact. In the treated hearts, the nanoparticles in increasing concentrations were also administered over 15 minutes via the cannula. Subsequently, the amplitudes of the contractions of the treated hearts were compared with the corresponding ones of the untreated time controls at the corresponding minutes of the recording. It is described in detail in the "Materials and Methods" Section, subsections 2.3. "Isolation of frog hearts and measurement of heart contractions". The clarification regarding the time control has been included in the same section of the revised manuscript, lines 189-190.

  1. In the three graphs of Figure 2, we only see these concentrations of 4, 10, 25, 50, and 100 μg/mL, and whether or not the authors compared these five concentrations to 0 μg/mL when comparing their results. In the graph, the author's arrow is pointing to the peak of the line graph, so are the authors comparing just the size of that peak? Or is it a segment of the folded graph. We don't think the folded graph presents the authors' results well.

Response: Thank you for your inquiry regarding the comparisons in Figure 2. To clarify, in our experiments, the control group constitutes samples untreated with any nanoparticles, effectively representing the 0 μg/mL group. The control group is denoted as the "Time control" group for reasons already explained in the answer to the above question. Consequently, the GO- and GO-PEG-treated groups are compared to the "Time control" group.

The arrows in figure 2 mark every change of the solution in the cannula, and an additional explanation has been added to the text under Figure 2. The changes can be seen in the revised version of the manuscript - lines 344-351

  1. In Line 331, The authors indicated that the severity, duration, and frequency of these injuries increased with increasing concentrations of applied nanoparticles. But in Figure 3, we do not see the above expression. We hope that the authors can complete the figure by adding exactly which particles and how many concentrations were exposed, so that the reading public can understand the authors' results more clearly.

Response: Thank you for your thoughtful feedback. We have considered your suggestion and addressed the concern regarding the alignment between the statement in Line 331 and the depiction in Figure 3. In response, we revised Figure 3 to include excerpts from the original data, illustrating the escalation of cardiac rhythm disturbances corresponding to the increasing nanoparticle concentrations. The revised figure and its associated text have been updated and integrated into the revised manuscript. We believe these revisions will significantly enhance the clarity and understanding of our results for the readers.

  1. In Line 383, what is the basis for the authors' choice of 5 minutes of NIR irradiation?

Response: The selection of a 5-minute duration for NIR irradiation in our study was based on preliminary experiments we conducted. We systematically tested various time intervals, including 1, 2, 5, and 10 minutes of irradiation. Surprisingly, our findings revealed that the 5-minute exposure had a more pronounced impact on cell viability than the 10-minute irradiation. This unexpected result prompted our choice of the 5-minute duration for further investigation, as it demonstrated a stronger effect on cellular viability in our preliminary assessments.

  1. In Figure 4, Line 385, we see in the description of the results that the authors compare the amount of ROS after GO and GO-PEG treatment. We think that in order to better present the authors' results, these four sets of data should be aggregated into a bar graph, if the authors want to compare them with each other.

Response: We appreciate this suggestion. To address it, we have combined two panels of Figure 4 into a single panel, and the text to the figure has been revised accordingly. The changes can be seen in the revised manuscript, around lines 380-386.

  1. In Line 380, the authors confirmed whether GO nanoparticles cause cytotoxicity by testing ROS levels, which we believe is incomplete. A complementary cell viability assay would have better interpreted the cytotoxicity caused by GO nanoparticles.

Response: Thank you for your comment. We recognize the significance of cell viability assays in thoroughly assessing nanoparticle-induced cytotoxicity, and we routinely include them in our cell culture experiments. In this specific study, however, our focus was on examining ROS levels to understand the impact of GO nanoparticles on mitochondria considering that the increased level of GO is a sign of mitochondrial damage. We intended to make a connection with other experiments in the study, more specifically with ATPase activity of rat liver mitochondria, to evaluate GO's effect on mitochondria using different experimental models and thus to understand the specific mechanisms of GO toxicity comprehensively. Your observation is indeed valid, and we appreciate your consideration.

  1. In the authors' figures, we can see that sometimes Control (Figure 5) is used to express the untreated group and sometimes 0 μg/mL (Figure 4) is used. If it is possible to standardize one form of presentation, it may make it easier for the reader to understand.

Response: Thank you for paying attention to this. All the figures have been revised, and untreated groups have now been labeled "Controls."

  1. In Figure 5, authors didn’t detect ATPase activity of GO-PEG treated. If this part is added to make the author's results more complete.

Response: We agree that exploring the impact of GO-PEG on rat liver ATPase activity would enhance the comprehensiveness of our study. Unfortunately, our initial experiments with these nanoparticles resulted in a cloudy solution in the samples for determining Pi levels, rendering the spectrophotometric measurements unreliable. The turbidity emerged at the outset of the procedure upon adding a reagent containing 0.162 mM ammonium molybdate and 1 M H2SO4 to the samples. Therefore, we presented only the results with GO nanoparticles here.

  1. The authors used frogs and rats as animal models respectively, why did the authors use two different animal models and on what basis?

Response: Our choice to use frogs and rats as animal models in the present research was guided by distinct scientific considerations. Each species represents a unique biological system, possessing specific physiological traits that offer advantages for different types of studies. For instance, frogs, with their simpler cardiovascular systems, were ideal for initial observations, providing clear insights into the direct effects of GO nanoparticles on the heart contractility. This aided in our assessment of the nanoparticles' cytotoxicity on the cardiovascular system. Conversely, with their physiological similarities to humans, rats offered a more complex biological system suited for in-depth studies. Wistar rats, widely utilized in studying human disorders and the effects of various substances, were employed in our research to assess the impact of GO nanoparticles on the activity of liver enzymes. Employing multiple models enables a comparative analysis, granting us insights into how diverse biological systems respond to these nanoparticles. This comparative approach bolsters the thoroughness of our study, enhancing our understanding of their potential effects on human health. By utilizing these different animal models, encompassing more straightforward and complex systems, our study aims to comprehensively evaluate the effects of graphene oxide nanoparticles from various biological perspectives. This multifaceted approach intends to draw more comprehensive conclusions regarding the potential impact of these nanoparticles on human physiology.

Thank you again for your feedback, and we hope these changes improve the quality of the manuscript.

Sincerely,

Prof. Natalia Krasteva

Assoc. Prof. Mariela Chichova

Round 2

Reviewer 1 Report

Comments and Suggestions for Authors

Accept.

Reviewer 3 Report

Comments and Suggestions for Authors

Accept